# Inhalation Anesthetics Play a Janus-Faced Role in Self-Renewal and Differentiation of Stem Cells

**DOI:** 10.3390/biom14091167

**Published:** 2024-09-18

**Authors:** Xiaotong Hao, Yuan Li, Hairong Gao, Zhilin Wang, Bo Fang

**Affiliations:** 1Department of Anesthesiology, The First Hospital of China Medical University, Shenyang 110001, China; 2023120907@cmu.edu.cn (X.H.); q111115492@gmail.com (Y.L.); 13941499242@163.com (H.G.); 2Department of Pain Medicine, The First Hospital of China Medical University, Shenyang 110001, China

**Keywords:** inhaled anesthetics, stem cells, self-renewal, proliferation, differentiation

## Abstract

Inhalation anesthesia stands as a pivotal modality within clinical anesthesia practices. Beyond its primary anesthetic effects, inhaled anesthetics have non-anesthetic effects, exerting bidirectional influences on the physiological state of the body and disease progression. These effects encompass impaired cognitive function, inhibition of embryonic development, influence on tumor progression, and so forth. For many years, inhaled anesthetics were viewed as inhibitors of stem cell fate regulation. However, there is now a growing appreciation that inhaled anesthetics promote stem cell biological functions and thus are now regarded as a double-edged sword affecting stem cell fate. In this review, the effects of inhaled anesthetics on self-renewal and differentiation of neural stem cells (NSCs), embryonic stem cells (ESCs), and cancer stem cells (CSCs) were summarized. The mechanisms of inhaled anesthetics involving cell cycle, metabolism, stemness, and niche of stem cells were also discussed. A comprehensive understanding of these effects will enhance our comprehension of how inhaled anesthetics impact the human body, thus promising breakthroughs in the development of novel strategies for innovative stem cell therapy approaches.

## 1. Introduction

Ether was initially employed as an anesthetic in 1842, marking the inception of the utilization of inhalation anesthetics in clinical settings as a customary agent for achieving general anesthesia [1]. These anesthetics are introduced into the lungs through inhalation via the respiratory tract, then infiltrating the bloodstream via the alveoli to permeate the central nervous system. Within the central nervous system, they foster unconsciousness, induce amnesia, impede pain signals, and prompt muscle relaxation. Inhalation anesthetics are distinguished by their rapid onset, manageable administration, and prompt, steadfast patient recovery. Consequently, inhaled general anesthesia remains a fundamental modality in contemporary anesthetic practice. Contemporary inhalation anesthetics include isoflurane (Iso), sevoflurane (Sev), etc. Inhalation anesthetics not only demonstrate anesthetic attributes, but also harbor deleterious effects on the organism. Maternal exposure to Sev during offspring neurogenesis impairs interneuronal migration and increases susceptibility to epilepsy in adolescence [2]. Clinical investigations have suggested that intraoperative Sev exposure can trigger postoperative delirium [3,4]. Mice exposed to Iso during pregnancy developed malformations in their offspring [5]. In addition, Sev enhances the proliferation and metastatic potential of cervical cancer cells [6]. However, the protective attributes of inhalation anesthetics towards the organism are also irrefutable. For example, Iso has been demonstrated to mitigate peripheral nerve damage in sciatic nerve repair procedures [7]. Sev has been shown to alleviate neurological damage resulting from cerebral ischemia–reperfusion and to inhibit the proliferation of breast cancer cells [8,9].

Stem cells, characterized by their undifferentiated nature, inhabit various developmental stages, including embryonic and adult environments. They possess a unique capacity for extensive proliferation and can differentiate into diverse cell types, thereby playing a foundational role in the formation of the fundamental units of tissues and organs [10]. Stem cells derive from multiple sources and serve distinct biological functions. Based on their differentiation capabilities, stem cells are categorized into specific types. Totipotent stem cells hold the potential to develop into complete organisms, akin to fertilized eggs. Pluripotent stem cells exhibit the ability to differentiate into a wide array of cell types, encompassing embryonic stem cells (ESCs) and mesenchymal stem cells. Unipotent stem cells possess the capacity to differentiate into a limited number of related cell types, exemplified by neural stem cells (NSCs) and hematopoietic stem cells [11]. Stem cells exhibit the ability to regenerate and repair damaged tissues, modulate the immune system, enhance blood circulation, eliminate aged and damaged cells, and facilitate cellular renewal [12,13,14,15]. As a special subset of cancer cells, cancer stem cells (CSCs) possess the characteristics of stem cells and are capable of driving the initiation, progression, metastasis, and recurrence of cancer [16].

Relationships between inhalation anesthetics and stem cells have been recently revealed. In view of the influence of inhaled anesthetic on the central nervous system, embryonic development, and tumor progression, NSCs, ESCs, and CSCs are closely related to the above pathophysiological processes. This review focuses on the above three types of stem cells. These findings provide valuable insights into the regulatory function of inhalation anesthetics on stem cells and reveal a promising new perspective for investigating the potential for preconditioning inhalation anesthetics before stem cell therapy.

## 2. NSCs

The impact of inhalation anesthetics on the nervous system has been a topic of ongoing research. The selection between Sev and intravenous anesthetics propofol did not appear to influence the prevalence of delayed neurocognitive recovery at 5–7 days post-surgery, as evidenced by the findings of the study [17]. In elderly patients undergoing major cancer surgeries, the incidence of delirium following Sev anesthesia is higher than that following propofol anesthesia [18]. Further study utilizing brain organoids has demonstrated that exposure to Sev temporarily increases neuronal differentiation but does not induce significant fetal brain developmental abnormalities in the long term [19]. Moreover, a substantial body of evidence from animal studies indicates that the administration of inhalation anesthetics may potentially induce neurotoxicity [20,21,22]. In light of these findings, it is evident that studies on different inhalation anesthetics under varying conditions have yielded contradictory conclusions. Accordingly, further investigation into the precise molecular mechanisms involved is imperative.

In 1992, Reynolds and Richards successfully isolated a population of cells from the striatum of adult mice that exhibited a remarkable capacity for self-renewal and differentiation into various cell types, such as neurons, oligodendrocytes, and astrocytes. These cells were identified as NSCs [23]. This discovery marked the commencement of focused research on NSCs. NSCs are predominantly localized in specific regions of the brain during the prenatal stage and persist in regions such as the subventricular zone in adult stages, the subgranular layer of the hippocampus, and the spinal cord. Their significance lies in their pivotal role in the genesis and maturation of the nervous system, influencing both normal neurodevelopment and the onset of neurological disorders. The transplantation of exogenous NSCs to supplement damaged nerve cells or activate endogenous NSCs for self-repair following injury holds significant clinical implications [24]. Investigations have unveiled that inhalation anesthetics possess the capacity to modulate the self-renewal and differentiation processes of NSCs, thereby potentially impacting either neurotoxic or neuroprotective effects on the brain [25,26,27,28].

### 2.1. Inhalation Anesthetics Affect NSCs in a Concentration and Time Dependent Manner

Recent works have revealed that the effects of inhalation anesthetics on NSCs are predominantly influenced by two key factors, although the underlying mechanisms are not fully understood. Firstly, the duration of exposure plays a critical role. Research findings have suggested that exposure to 4.1% Sev for less than 4 h does not elicit considerable harm to NSCs. However, prolonged exposure exceeding 8 h has been associated with the promotion of apoptosis, diminished proliferative capacity, reduced differentiation potential of NSCs, and an upsurge in the population of fully differentiated cells [29]. Nevertheless, diverging scholarly opinions exist, with some scholars contending that the proliferation of NSCs experiences an increase up to a certain threshold subsequent to exposure to Sev concentrations of 2.4% or 3.6% for 1 h. Interestingly, a 6 h exposure period has shown elevated inhibition of proliferation and increased apoptosis, yet the level of differentiation remains unaltered across varying doses and exposure durations [30]. Secondly, the concentration of exposure emerges as another crucial determinant. Iso did not demonstrate a notable antiproliferative effect when NSCs were treated with the next-lowest alveolar concentration (0.7%) of Iso in comparison to higher alveolar concentrations (1.4% and 2.8%) [31]. The results of animal studies have suggested that in contrast to exposure to subclinical concentrations (1.2%) of Sev, exposure to higher doses (2.4%, 4.8%) inhibited proliferation and promoted apoptotic cell death and cell differentiation [32]. These investigations underscore the bidirectional effects of inhalation anesthetics on NSCs. While the concentration and duration of drug exposure are pivotal factors, the precise threshold where protective effects transition into toxic consequences remains indeterminate. Additionally, it is important to acknowledge that the sources of these NSCs and their cultural conditions are not entirely uniform, and these subtle differences may also influence the conclusions drawn from the experiments. Therefore, further rigorous and comparable research is needed to address the issues related to concentration and time dependency (Table 1).

### 2.2. Effects of Inhalation Anesthetics on NSCs In Vivo

Some in vivo experiments have yielded results consistent with those observed in vitro. Pregnant rats were subjected to various concentrations of Sev, with subsequent collection of fetal brain tissue samples for analysis. In mid-gestation rats, Sev has been demonstrated to inhibit the proliferation of fetal NSCs via the Wnt/β-catenin pathway [33]. However, certain studies have indicated that NSCs exhibit varying responses to inhalation anesthetics in both in vivo and in vitro settings. While the diminished proliferation capacity of NSCs in vitro can recover within 24 h post-exposure to 0.7% Iso, the effects on NSCs in vivo may persist for days or even weeks. Iso induces proliferation inhibition by causing cell cycle arrest directly in NSCs. The enduring anti-proliferative effects observed in vivo may be due to the secondary impacts of Iso on peripheral nervous tissue [31]. Additionally, Nie et al. noted that brief exposure (1 h) to clinical levels of Sev enhanced NSCs proliferation in vitro, yet did not stimulate neurogenesis in vivo [30]. Discrepancies between in vivo and in vitro outcomes suggest that the regulation of stem cell proliferation in vivo is markedly more complex. It is reported that the fate of NSCs is precisely controlled by alterations in their microenvironment, referred to as the niche. The self-renewal and differentiation of NSCs rely on their interactions within this microenvironment. For instance, cerebrospinal fluid fosters NSCs proliferation in the subventricular zone by way of epithelial sodium channels, while angiogenesis in the niche is intricately linked to neurogenesis [34]. Following exposure to inhaled anesthetics, alterations may occur in neurotransmitters, growth factors, and membrane-associated ligands [35,36]. These changes may collectively affect the behavior of NSCs. Actually, in the clinical setting, the mechanisms are often intertwined. Hence, further research is necessary to accurately delineate the effects of inhalation anesthetics on NSCs in vivo.

The amplification of NSCs within the hippocampus is a fundamental requirement for the process of synaptogenesis, a critical stage in the maturation of hippocampal learning and memory networks [37]. In the context of cognitive function in rats, exposure to inhalation anesthetics has been widely reported to induce impairment, with close connections to NSCs. Specifically, research has illustrated that exposure to Sev during the second trimester of pregnancy led to diminished learning and memory capacities in offspring rats, showing a dose-dependent effect. This effect was attributed to the hindrance of hippocampal NSCs proliferation, alterations in neurogenesis within the hippocampus, and disruption of neural circuitry formation [33]. Subsequent investigation revealed a notable reduction in the proliferation and differentiation potential of NSCs following a two-week exposure to either Iso or Sev. Assessments using the Morris water maze test demonstrated a cognitive decline in the tested rats during this period, with subsequent recovery observed within six weeks post-anesthesia [38]. The precise dosage threshold for cognitive impairment induced by inhalation anesthetics is inconclusive, underlining the necessity for further comprehensive investigations to inform clinical therapeutic strategies.

## 3. ESCs

In recent years, more and more pregnant women have been exposed to inhaled anesthesia, and the effects of inhaled anesthesia on pregnant women and fetuses have aroused wide concern. The potential implications of this issue involve maternal health, fetal development, and the safety of the delivery process. A cross-sectional study examining the effects of inhalation anesthetics exposure on reproductive outcomes among healthcare workers at Jimma University Hospital found that those exposed to inhalation anesthetics had a higher prevalence of adverse reproductive outcomes such as spontaneous abortion and preterm birth [39]. In contrast, other studies have shown no statistically significant differences in miscarriage and congenital malformation risks among women working in veterinary anesthesia compared to unexposed groups [40]. Moreover, it was found that mice exposed to Sev during pregnancy exhibited hearing impairments in their offspring [41]. These findings inevitably lead us to consider the relationship between inhalation anesthetics and ESCs.

ESCs originate from the inner cell mass of developing blastocysts and exhibit distinctive traits, including unlimited proliferation, self-renewal capacity, and the ability to differentiate into various cell types in vitro [42]. When subjected to specific environmental cues, ESCs possess the capability to develop into cells representing all three germ layers in vivo. Inhalation anesthetics can cross the placental barrier and the blood–brain barrier, producing harmful effects on ESCs.

In vitro experiments, treating ESCs with 4.1% Sev for 4 or 6 h resulted in increased apoptosis, reduced self-renewal capacity, and delayed differentiation compared to control cells, with these results being time-dependent [42,43]. Iso treatment also demonstrated similar outcomes. When ESCs were exposed to 2% Iso for 2, 4, and 6 h, a significant decrease in the size of the embryoid bodies produced by the ESCs was noted, along with a marked reduction in their proliferation capacity, leading to decreased neuronal differentiation [44,45]. In vivo experiments further validated these conclusions. Pregnant mice were treated daily with 1.4% Iso for 2 h over three consecutive days during gestation days E3.5 to E6.5, simulating the exposure conditions experienced by pregnant women undergoing prolonged non-obstetric surgeries. At E18, the pregnant mice were euthanized, and the weights of their pups were measured, revealing impaired fetal growth and development [44,45]. Yi et al. applied a higher concentration of 3% Sev under the same conditions on pregnant mice and found a significant inhibition in the expression of genes associated with fetal brain development [42]. It can thus be concluded that inhalation anesthetics not only inhibit the self-renewal capacity and differentiation of ESCs at high concentrations, but also exert toxic effects on ESCs under prolonged treatment at concentrations approaching those used clinically. Importantly, this conclusion has been validated in both in vivo and in vitro studies.

Inhalation anesthetics are thought to be harmful rather than protective to ESCs, in contrast to the bidirectional effect of inhalation anesthetics on NSCs. It is crucial to investigate effective strategies for mitigating the adverse effects of inhaled anesthetics on embryonic development, including abortion and developmental delay. Furthermore, additional research is needed to determine whether inhalation anesthetics could potentially exhibit any protective effects on ESCs. This will provide important insights for developing safer and more effective treatment strategies for anesthesia during pregnancy.

## 4. CSCs

Recent studies increasingly suggest that perioperative administration of anesthetics significantly influences cancer recurrence, metastasis, and ultimately, long-term survival outcomes in patients [46,47]. For example, in breast cancer patients, cancer recurrence rates after modified radical mastectomy were higher with inhalation anesthesia than with intravenous propofol anesthesia [47]. Similarly, in laparoscopic hepatectomy, the use of inhalation anesthetics as the primary general anesthetic compared with intravenous propofol may result in an increased 2-year recurrence rate for early and intermediate hepatocellular carcinoma [46]. Sev and desflurane have been demonstrated to inhibit the expression of miR-138 and miR-210, which increases the proliferation and migration of ovarian cancer cells [48]. Inhaled anesthetics, however, have a complex dual effect on cancer behavior, and many studies have shown that inhaled anesthetics possess anticancer properties. For instance, the result demonstrated that Sev led to a reduction in the proliferation and invasiveness of neuroblastoma cells, as well as an increase in apoptosis. This resulted in a delay in cancer growth [49]. Moreover, Sev has been demonstrated to downregulate Akt1 expression levels in a dose-dependent manner, thereby inhibiting glioma cell migration and invasion [50]. These contradictory findings have spurred further research into the underlying mechanisms. CSCs constitute a subset of cancer cells within malignant tissues. They exhibit pluripotency, enabling self-renewal and differentiation into various cancer lineages [51]. CSCs undergo asymmetric division, generating two daughter cells with distinct destinies: one retaining stem cell characteristics for self-renewal, while the other transforms into a specialized progenitor cell capable of producing mature cancer cells, contributing extensively to cancer mass [52]. A growing body of evidence from recent studies has begun to elucidate the effects of inhaled anesthetics on CSCs and the underlying mechanisms.

Treatment of glioma stem cells (GSCs) derived from high-grade glioma patients with 2% Sev (the minimum alveolar concentration during anesthesia in a clinical setting at the start of surgery) does not affect cell apoptosis, proliferation capacity, or colony formation ability, even after prolonged exposure (4–6 days) [53]. It has been reported that human glioma cell lines U87MG and U373MG can give rise to GSCs. Han et al. exposed human glioma cells to Sev at concentrations of 1.0%, 2.5%, and 5% for 24 h, demonstrating that Sev inhibits the stemness of GSCs in a dose-dependent manner, thereby exerting anti-cancer effects [54]. However, exposure of primary human GSCs to 2% Sev for 6 h resulted in a significant increase in actively proliferating GSCs, as well as enhanced sphere-forming ability of isolated single cells [55]. Additionally, GSCs exposed to various concentrations of Iso over different time periods exhibited increased proliferation activity, improved survival rates, and enhanced growth potential compared to control conditions. Moreover, GSCs previously exposed to Iso demonstrated advanced migratory capabilities both in vitro and in vivo. This finding suggests that Iso has a dose-dependent and time-dependent ability to promote the proliferation, survival, and migratory potential of human glioblastoma stem cells, potentially contributing to the pathogenesis of glioblastoma [56].

Both clinical and basic studies have found a variety of effects of inhalation anesthetics on cancer or CSCs. The effects of inhalation anesthetics on CSCs may be influenced by numerous factors, including the concentration of inhaled anesthetics, the duration of treatment, and variations in the source of cells utilized. To obtain more precise and dependable conclusions, further research is essential to facilitate the judicious use of inhaled anesthetics for more effective inhibition of cancer progression and recurrence.

## 5. Mechanisms of Inhalation Anesthetics on Stem Cells

### 5.1. Cell Cycle

The process of cell division plays a critical role in the self-renewal and differentiation of stem cells, which is regulated by mechanisms within the cell cycle operating in the nucleus. The cell cycle comprises four phases: Gap 1 (G1), DNA Synthesis (S), Gap 2 (G2), and Mitosis (M) [57]. Progression through these stages is predominantly driven and regulated by two classes of proteins: cyclin-dependent kinases (Cdks) and cyclins [58,59]. Cyclins activate Cdks by forming complexes known as cyclin–Cdk complexes [60]. The progression of the cell cycle is influenced by signaling pathways initiated by growth factors that modulate the activity of various Cdk–cyclin complexes.

The Wnt signaling pathway is essential for maintaining a balance between the proliferation and differentiation of stem cells [61]. Ruan et al. reported that Sev exerts inhibitory effects on the growth, differentiation, and self-renewal of chronic myeloid leukemia (CML) CD34+ stem cells through suppression of the Wnt/β-catenin signaling pathway [62]. Key components of this pathway include glycogen synthase kinase 3β (GSK-3β) and β-catenin. In this signaling cascade, GSK-3β phosphorylates β-catenin, leading to its proteasomal degradation. Activation of the Wnt pathway inhibits GSK-3β, resulting in the accumulation of β-catenin in the cytoplasm. Subsequently, β-catenin translocates to the nucleus, where it acts as a transcription factor to activate downstream target genes that regulate the transcription of G0/G1 cell cycle regulators [63]. Research conducted by Liu and Wang et al. on the proliferation and differentiation of NSCs following exposure to Sev, both in vitro and in vivo, revealed that high concentrations of Sev upregulate the expression of GSK-3βand promote the degradation of β-catenin. This interference results in reduced expression of cell cycle regulators such as CD44 and Cyclin D1, ultimately leading to cell cycle arrest at the G0/G1 phase. Prior studies have indicated that a shortened G1 phase in neuroprogenitor cells inhibits the transition from proliferation to differentiation, while an extended G1 phase facilitates this transition [64,65]. Consequently, exposure to Sev appears to prolong the G0/G1 phase, inhibiting the proliferation of the NSC population and promoting differentiation [33,66,67].

p53 functions as a transcription factor that activates a wide array of genes, ultimately inducing apoptosis and cell cycle arrest [68]. p21 is one of the transcriptional targets of p53 and mediates p53-induced G1 cell cycle arrest [69]. Following treatment with varying concentrations of Iso, the liver kinase B1(LKB1) phosphorylates the p53 transcription factor at specific sites, enhancing its affinity for its target promoter, the p21 gene. Increased expression of p21 subsequently affects the formation of Cdks and cyclin complexes, leading to cell cycle arrest [70,71].

Furthermore, Chen et al. discovered that overexpression of Hsp70 in NSCs can prevent the decline in cognitive abilities observed in offspring mice exposed to Sev during pregnancy [72]. Previous studies have shown that Hsp70 can activate extracellular signal-regulated kinase (ERK)-related signaling pathways, which are critical in regulating cell proliferation and differentiation [73]. Activated mitogen-activated protein kinase (MAPK) -ERK can induce CyclinD expression and promote cell G1/S transition [74,75]. Therefore, it is plausible to hypothesize that Hsp70 modulates the cell cycle in NSCs through the activation of ERK-related signaling pathways, thereby mitigating the toxic effects of Sev on these cells. Gamma-aminobutyric acid (GABA) is an inhibitory neurotransmitter present in various cell types within the central nervous system. It plays a critical role in regulating cellular proliferation, differentiation, immune modulation, and other physiological processes [76,77,78]. With 4.1% Sev treatment of ESCs cells for 6 h, p-ERK phosphorylation level increased, cell cycle arrest in S phase occurred, self-renewal was inhibited, stem cell gene expression level decreased, proliferation slowed down, and apoptosis increased. Knockdown of γ-aminobutyric acid A receptor (GABA_A_R) attenuates Sev promotion of p-ERK phosphorylation and rescues ESCs from Sev effects [43]. Activation of ERK is generally thought to contribute to cell proliferation [79]; however, recent studies have shown that ERK may also have a pro-apoptotic effect [16]. Different cell types and different processing conditions may be responsible for this contradictory result.

The study of the potential effects of inhalation anesthetics on stem cell cycle regulation offers valuable insights, particularly with regard to signaling pathways and gene expression associated with regulating different stages of the cell cycle. These insights will inform the development of strategies to optimize stem cell therapy in clinical settings (Figure 1).

### 5.2. Metabolism

Metabolism plays a crucial role in the regulation of stem cell fate. Recent studies have shown that cellular metabolism is not only the basis for the energy and material requirements of stem cells and their differentiated progeny, but also an important determinant influencing signaling pathways, chromatin modifications, and gene expression [80,81,82]. Particularly under different physiological conditions, the metabolic networks within cells undergo dynamic adjustments to meet specific metabolic demands, thereby affecting stem cell self-renewal, differentiation, and apoptosis [83,84].

Among various metabolic regulatory pathways, the Akt-GSK-3β signaling pathway is considered a key metabolic regulator [85]. Akt (protein kinase B) participates in regulating cellular energy metabolism and autophagy by inhibiting the activity of p-GSK-3β [86,87,88]. Furthermore, this pathway also plays a role in neuroprotection by preventing mitochondrial-dependent neuronal apoptosis, thus maintaining cell survival [89]. Under the influence of inhalation anesthetics, the Akt-GSK-3β pathway demonstrates complex regulatory effects. High concentrations of Iso can lead to decreased expression of phosphorylated Akt and GSK-3β and increased cleaved caspase-3. These changes ultimately initiate a decline in the proliferation and survival of NSCs, increasing the rate of apoptosis and resulting in cognitive impairment and neurotoxicity [90].

Intracellular calcium (Ca^2+^) signaling generated from the endoplasmic reticulum influences several physiological processes, including the regulation of mitochondrial homeostasis and determining cell survival and death [91,92]. Sev may disrupt Ca^2+^ homeostasis by elevating Ca^2+^ concentration, leading to endoplasmic reticulum stress, which ultimately activates CaMKII and stimulates the phosphorylation of CaMKII and c-Jun N-terminal kinase (JNK) in a dose-dependent manner, inhibiting tumor growth and the stemness of GSCs both in vitro and in vivo. Additionally, after treatment with Sev, GSCs exhibit significantly decreased levels of intracellular and mitochondrial reactive oxygen species (ROS), as well as diminished mitochondrial membrane potential [54]. These findings reveal the inhibitory effects of Sev on mitochondrial function in GSCs, while mitochondrial metabolism and ROS generation are crucial for cell proliferation and tumorigenesis [93,94]. Therefore, these results underscore the potential impact of Sev in regulating GSCs’ functions.

Similarly, long non-coding RNAs (lncRNAs) can regulate ROS production to influence cell growth [95]. Lu and colleagues demonstrated that the downregulation of lncRNA Wnt5A-AS may inhibit Wnt5A transcription factor activity, leading to reduced Wnt5A expression and concomitant suppression of RYK expression. Furthermore, they showed that the reduction in Wnt5A and RYK together results in decreased ROS levels and facilitates the entry of quiescent NSCs into the cell cycle. This process ultimately enhances the proliferation of NSCs exposed to Sev [96].

Iron, as an essential trace element, participates in various metabolic processes within the central nervous system, including energy production, DNA synthesis, and oxygen transport [97]. Research has shown that maternal exposure to Sev induces disturbances in iron metabolism in the fetal mouse brain, impacting postnatal cognitive function. Following exposure to high concentrations of Sev, the expression of iron transporter 1 (FpN1) decreases, resulting in reduced intestinal iron absorption and subsequent brain iron deficiency. Concurrently, there was an up-regulation of iron transporter 1 (TfR1) and a decrease in ferritin expression, which led to functional impairments in iron metabolism and significantly suppressed the proliferation of NSCs [98].

Adenine nucleotide translocase 1 (ANT1), an ADP/ATP translocase, also promotes mitochondrial autophagy [99,100]. It is commonly assumed that ANT1 facilitates the occurrence of oxidative phosphorylation, which is the primary metabolic process during the differentiation of NSCs [101,102]. However, following Sev treatment, the upregulation of miR-410-3p directly affects ANT1, causing a substantial decrease in its levels, which unexpectedly promotes the premature differentiation of NSCs [103]. Future studies should aim to explore the different molecular mechanisms of ANT1’s dual roles more deeply.

Understanding how anesthetics influence these key metabolic pathways provides new perspectives for developing intervention strategies aimed at mitigating anesthesia-related cytotoxicity and protecting the proliferation and survival of stem cells. Future research in this field could focus on developing drugs targeting specific metabolic pathways or combining nutritional interventions to optimize stem cell function (Figure 2).

### 5.3. Stemness

The regulation of stem cell properties, particularly “stemness”, is essential for maintaining a delicate balance between self-renewal and differentiation. Stemness encompasses the intrinsic capabilities of stem cells, including their ability to maintain self-renewal and differentiate into multiple lineages while preventing premature differentiation. This dynamic characteristic is influenced by various intrinsic and extrinsic factors, such as cell size, cellular state, and the surrounding microenvironment [104,105]. Recent research has emphasized the effects of anesthetic agents on the maintenance of stemness in stem cells, suggesting potential implications for developmental outcomes following exposure. Sex-determining region Y-box 2(Sox2), Octamer-binding transcription factor 4 (Oct4), Nanog and Kruppel-like factor 4 (Klf4) are key factors involved in the process of cellular reprogramming and serve as markers of stemness; they play critical roles in sustaining the pluripotency of stem cells [106]. Inhalation anesthetics affect the expression of these factors and other stemness-related molecules, thereby regulating cell stemness.

Research has shown that Sev induces the upregulation of miR-183, which inhibits the expression of Sox2, subsequently suppressing the proliferation and differentiation of NSCs [107]. Lin28 is an RNA-binding protein that plays a significant role in early embryonic development, stem cell differentiation, and reprogramming [108]. Following the exposure of early pregnant mice to Sev, the upregulation of let-7a in ESCs resulted in the downregulation of Lin-28a levels, a reduction in Sox2 levels, and a notable impairment in the differentiation capacity of ESCs, in addition to an impact on self-renewal [42]. Sox2 belongs to the SRY-related HMG-box (Sox) family of transcription factors, which also includes Sox13, a factor closely associated with the proliferation and differentiation of NSCs [109,110]. Repeated exposure to Sev results in reduced expression of LncRNA-Peg13 in NSCs, leading to elevated levels of miR-128-3p and decreased expression of Sox13, ultimately exerting negative effects on the self-renewal and differentiation of embryonic NSCs in mice [111].

Nanog and Oct4 are core transcription factors involved in maintaining stem cell pluripotency [112,113]. Various molecules influence stem cell stemness by regulating the expression of Oct4 and Nanog. Retinoic acid receptor gamma (RAR-γ) plays a critical role in balancing the processes of stem cell self-renewal and differentiation [114,115]. Low expression or loss of RAR-γ results in a decrease in the self-renewal capability of stem cells, accompanied by an increase in differentiation. Research has shown that Iso inhibits the transcription of stemness factors Nanog and Oct4 by downregulating RAR-γ, thus hindering the self-renewal of mouse ESCs [44]. Furthermore, JNK as a member of the MAPK family, critically regulates the stemness of various stem cells [116]. Experimental results indicate that exposure to Sev significantly increases the expression of p-JNK while decreasing the proportion of proliferating and undifferentiated NSCs. Inhibiting JNK gene expression substantially lowers the levels of Nanog and Oct4 [117,118]. These findings suggest that Sev may trigger JNK-mediated damage in NSCs, reducing their stemness [32]. Additionally, it has been discovered that maternal exposure to Sev leads to premature differentiation of NSCs into neurons and astrocytes in the fetal brain, accompanied by upregulation of NRF2, ultimately resulting in a reduced number of neurons and increased proliferation of astrocytes in the hippocampus of postnatal rats [119]. Further studies have shown that NRF2 can promote the differentiation of rat NSCs via the SHH/GLI1 signaling pathway. Sonic Hedgehog (SHH), a member of the HH family, can directly regulate GLI1 expression, thereby promoting the expression of Nanog and Oct4 and consequently modulating stem cell stemness [120,121]. Thus, further research is needed to determine whether inhalation anesthetics can impact stem cell stemness through the NRF2/SHH/GLI1 signaling pathway.

Klf4 is an evolutionarily conserved zinc-finger transcription factor and unquestionably one of the important stem-related transcription factors. Many signaling pathways regulate the expression and function of Klf4, playing a crucial role in cell cycle regulation, somatic cell programming, and pluripotency [122,123,124]. Studies have definitively shown that knocking out E-cadherin results in a loss of cell adhesion in stem cells and impaired expression of the pluripotency-related transcription factor Klf4 [125,126]. Furthermore, E-cadherin is expressed on a variety of stem cells and interacts with E-cadherin molecules in neighboring cells to facilitate homologous cell-to-cell contact. This plays an important role in the induction of pluripotent stem cells [127]. It is noteworthy that Iso exposure elevates miR-9 in ESCs, which directly targets and inhibits E-cadherin, effectively blocking ESC self-renewal, stem gene expression, and neuronal differentiation [45].

In light of the evidence pertaining to the effects of inhalation anesthetics on stem cell properties, it is imperative that future research endeavors to elucidate the underlying molecular mechanisms involved. An understanding of the mechanisms by which inhalation anesthetics regulate stem cell stemness could provide significant insights into the development of stem cell therapy (Figure 3).

### 5.4. Niche

Precise control of stem cell self-renewal and differentiation is essential for proper organogenesis and tissue homeostasis. Alongside stem cells exists a specialized microenvironment known as the “niche”. The normal niche consists of fibroblasts, immune cells, endothelial cells, perivascular cells or their progenitors, extracellular matrix (ECM) components, and a variety of cytokines and growth factors [128]. These factors work together to determine the fate of stem cells under specific physiological or pathological conditions.

In the context of neuroinflammation, released cytokines and chemokines exert inhibitory effects on neurogenesis [129]. For instance, pro-inflammatory cytokines such as TNF-α and IL-6 secreted by microglia can exacerbate neurotoxicity, suppress NSCs proliferation, and promote their differentiation into astrocytes [130,131,132]. Studies have shown that Iso and Sev induce neurodevelopmental toxicity by regulating microglial activation and increasing levels of pro-inflammatory cytokines (e.g., IL-6 and TNF-α), resulting in learning and memory impairments in neonatal rats [38]. However, microglia are also capable of releasing factors that support NSCs proliferation and maintenance, indicating their dual role in regulating stem cell behavior [133].

Macrophages similarly have a significant impact on stem cell behavior [134]. Among them, tumor-associated macrophages (TAMs) are important tumor-infiltrating immune cells that secrete numerous factors, including epithelial growth factor (EGF), platelet-derived growth factor (PDGF), TGF-β1, and more. These factors regulate the tumor microenvironment and affect the function of CSCs [135]. It has been postulated that Sev may exert beneficial effects by reducing TAM within the tumor microenvironment, thereby rendering the tumor susceptible to cytotoxic T cells and immunotherapy [136]. Additionally, Iso and Sev have been demonstrated to attenuate macrophage phagocytosis, regulate macrophage polarization, and improve cell viability [137,138,139]. However, the precise manner in which they affect macrophage function and regulate stem cell behavior remains to be elucidated.

The ECM provides structural and biochemical support for stem cells, and inhalation anesthetics have the potential to affect ECM function [140]. For example, Sev has been demonstrated to downregulate EGF-containing fibrin extracellular matrix protein levels and EGF-containing extracellular matrix protein 2 (Fibulin-4) levels, thereby promoting apoptosis [141]. As ECM components are essential for delivering the external signals required to maintain stemness, any alteration can markedly influence stem cell behavior.

A more profound examination of the interactions between stem cells and their niches will not only elucidate the enigmas of stem cell biology in vivo but also offer novel therapeutic prospects for disciplines such as neurodegenerative diseases and oncology. In light of the intricate nature of stem cell niches, there is a pressing need to develop more sophisticated model systems to effectively model and analyze the effects of niches on stem cell biological processes (Figure 4).

## 6. Conclusions and Future Perspectives

In recent decades, an accumulating body of literature has not only provided compelling evidence of the effects of inhalation anesthetics on stem cells, but also greatly expanded our knowledge of the cellular and molecular basis of this effect. In this review, we have presented an extensive discussion of how inhalation anesthetics can regulate the proliferation and differentiation of NSCs, ESCs, and CSCs via various mechanisms. However, there are still some limitations in the current research.

Numerous studies have investigated the effects of inhalation anesthetics on stem cells and their underlying mechanisms through animal models and cell experiments. However, due to safety and technical limitations, few clinical studies examining the effects of inhaled anesthetics on human stem cells have been reported. With the continuous advancement of new technologies such as in vivo imaging and single-cell sequencing, there is hope for conducting more clinical studies on humans in the near future. These studies aim to provide direct evidence validating the effects of inhalational anesthetics on human stem cells. In this review, we mainly discussed isoflurane and sevoflurane, which are commonly used in clinics and widely studied. Xenon, as a novel inhalational anesthetic, has been demonstrated to be safer and to possess cytoprotective properties, providing therapeutic benefits in neuroprotection and organ transplantation preservation [142,143,144]. Thus, it is evident that further investigation into the effects of xenon on stem cells presents a promising avenue for research.

Previous studies investigating the effects of inhalation anesthetics on stem cells have concentrated on the biological processes of proliferation and differentiation. In recent years, the studies of inhaled anesthetics on stem cell migration have received much attention [145,146]. Cell migration is fundamental for multicellular organisms to establish and maintain normal tissue [147]. The inherent migratory capacity of stem cells aids in maintaining tissue homeostasis and promoting repair and regeneration [148]. Extracellular vesicles are a class of cell-derived membrane structures, including exosomes and microvesicles, involved in various physiological and pathological processes. Extracellular vesicles are now recognized as mechanisms of intercellular communication, allowing cells to exchange proteins, lipids, and genetic material [149]. Stem cells secrete protective factors through extracellular vesicles to create a regenerative microenvironment that facilitates tissue maintenance and repair processes [150]. However, the effects of inhalation anesthesia on extracellular vesicles of stem cells have not been reported. In the future, priority should be given to extracellular vesicle secretion of stem cell, which is expected to make significant progress [151].

Moreover, the effects of inhalation anesthetics on stem cells provide new perspectives for clinical applications, especially in the realm of regenerative medicine. Regenerative medical technologies centered around stem cells hold immense potential for improving human diseases, delaying aging, and enabling tissue regeneration [152,153,154]. One of the major challenges facing stem cell transplantation is immune rejection, which leads to the transient survival of transplanted stem cells. Preconditioning with inhalation anesthetics prior to transplantation might enhance the survival and self-renewal of stem cells in vivo. Additionally, the primary methods of stem cell transplantation involve non-invasive blood transfusion and invasive targeted injections, with the latter requiring surgical intervention [155]. Prior research has demonstrated that inhalation anesthetics can mitigate the perioperative immune response via a range of molecular mechanisms [156]. It would be of interest to investigate whether the appropriate selection of inhalation anesthetics during surgery can affect the microenvironment of stem cells, reduce immune rejection, and improve the survival rate of transplanted stem cells.

We also observe that the application of the same inhalational anesthetic can lead to neuroprotection in pediatric patients while causing neuronal damage in the elderly [157,158]. Animal studies have demonstrated that under identical conditions, treatment with Iso results in memory impairments exclusively in juvenile rats and not in adult ones [159]. This difference suggests that inhaled anesthetics may play different roles at different ages. Furthermore, the impact of inhalation anesthetics may vary among different types of cancer. For instance, Sev has been observed to inhibit the self-renewal of CD34+ stem/progenitor cells in chronic myeloid leukemia, while simultaneously promoting the proliferation of GSCs [55,62]. Additionally, the same inhalational anesthetic may elicit markedly different responses in various cell types. For example, Sev can enhance the proliferation of GSCs while suppressing the proliferation of NSCs [54,107]. These discrepancies may arise from variations in underlying mechanisms. With the rapid advancement of high-throughput sequencing technologies, we can develop strategies to compare molecular expression differences among various stem cells treated with inhalation anesthetics, thereby analyzing the specific mechanisms underlying their differential effects. This could facilitate a better understanding of how inhalation anesthetics operate across different stem cell types. What is intriguing is whether the mechanism by which inhaled anesthetic inhibits the proliferation of NSCs is applicable to the inhibition of tumor progression by CSCs. Conversely, does the mechanism that promotes GSCs proliferation also support NSCs self-renewal? Therefore, determining how to apply tailored treatment protocols for different patient populations while ensuring safety to maximize the therapeutic potential of inhalation anesthetics remains an area worthy of further investigation.

Taken together, further research is required to elucidate the long-term effects of inhalation anesthetics on stem cells and the precise regulation of stem cell behavior. These findings could help enhance precision anesthesia management and explore other clinical applications of inhaled anesthetics.

## Figures and Tables

**Figure 1 biomolecules-14-01167-f001:**
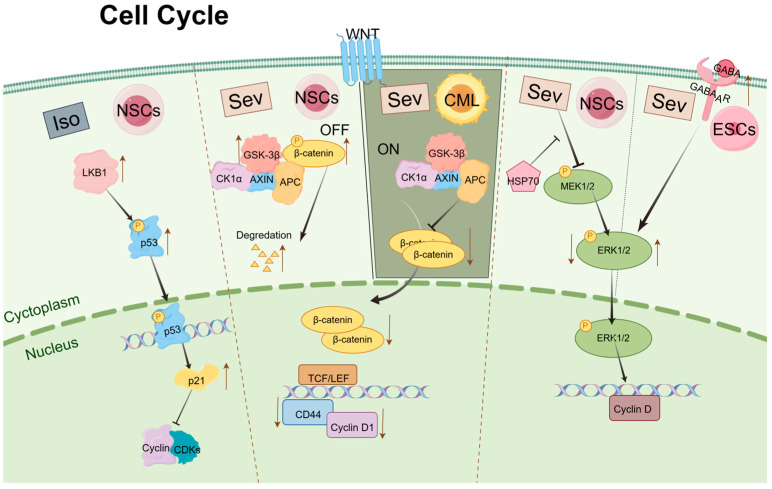
Mechanisms of Sev and Iso regulating cell cycle in stem cells.

**Figure 2 biomolecules-14-01167-f002:**
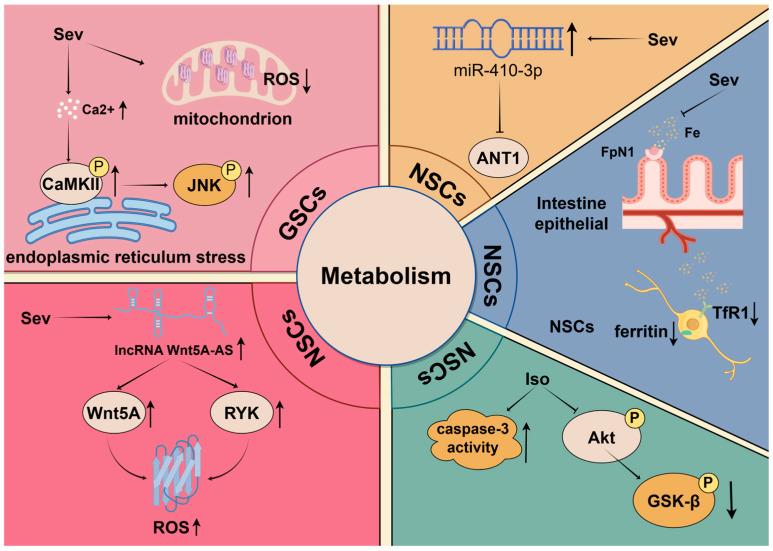
Mechanisms of Sev and Iso regulating metabolism in stem cells.

**Figure 3 biomolecules-14-01167-f003:**
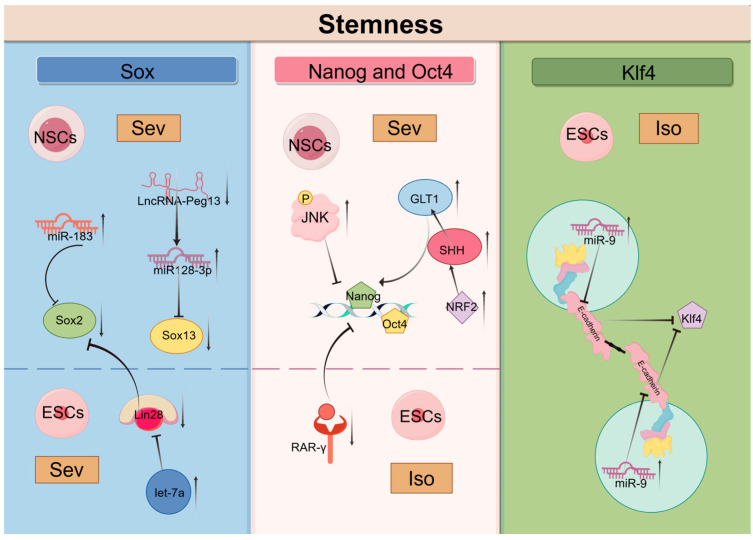
Mechanisms of Sev and Iso regulating stemness in stem cells.

**Figure 4 biomolecules-14-01167-f004:**
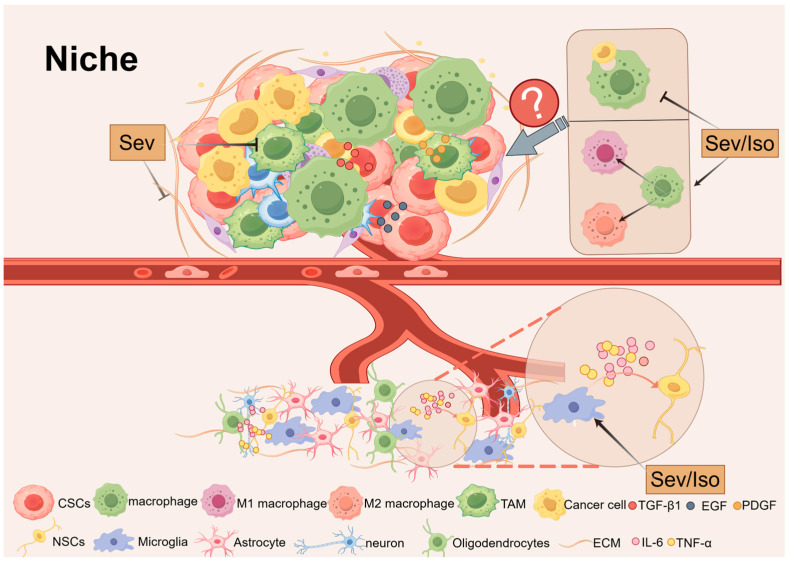
Mechanisms of Sev and Iso regulating niche in stem cells.

**Table 1 biomolecules-14-01167-t001:** The impact of inhalation anesthetics on NSCs varies based on concentration and duration.

Regional Sources	Stage of Sources	Inhalation Anesthetics	Exposure Concentration	Time	Effects	References
Forebrain lateral ventricle	Embryo (E15)	Sev	4.1%	4 h	-	[29]
				8 h	Promoted apoptosis, reduced proliferative and differentiation	
Hippocampi	Embryo (E14.5–16.5)	Sev	1.2%	1 h	-	[30]
			2.4%		Increased proliferation and viability	
			3.6%			
			1.2%	6 h	-	
			2.4%		Reduced proliferation and promoted apoptosis	
			3.6%			
Cortices	Embryo (E14)	Iso	0.7%	6 h	-	[31]
			1.4%		Reduced proliferation	
			2.8%			
Cortices	Embryo (E14)	Sev	1.2%	6 h	-	[32]
			2.4%		Inhibited proliferation, increased apoptotic cell death, and promoted cell differentiation	
			4.8%			

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
