# Peer review of "Inhalation Anesthetics Play a Janus-Faced Role in Self-Renewal and Differentiation of Stem Cells"

_biomolecules, 2024, doi:10.3390/biom14091167_

Round 1
Reviewer 1 Report
Comments and Suggestions for Authors
This review does not provide new insights or a fresh perspective on the impact of inhaled anaesthetics on stem cells. In fact literature is plenty of papers on this very controversial topic.
The authors primarily summarise existing literature without offering a unique angle or novel hypotheses, which is crucial for a review paper to contribute meaningfully to the field.
The discussion does not delve deeply enough into the mechanisms by which anesthetics affect stem cells.
While the paper covers various signaling pathways and molecular interactions, it often stops short of critically evaluating conflicting findings or identifying gaps in current knowledge.
There seems to be a lack of emphasis on the most recent advances and emerging trends in the field. A comprehensive review should highlight the latest studies and how they contribute to or challenge existing paradigms.
The review does not adequately address the limitations of the studies it cites, nor does it discuss the controversies or debates within the field. A critical evaluation of methodologies and findings is necessary to provide a balanced and informative overview.
The manuscript focuses extensively on certain signaling pathways, such as the Akt-GSK-3β pathway, potentially overlooking other important mechanisms or broader implications. A more balanced discussion covering a wider range of molecular mechanisms would provide a more holistic view.
While the paper mentions the implications for personalized medicine, it does not sufficiently explore how the findings could be translated into clinical practice. Discussions on potential therapeutic applications, safety concerns, and future research directions are somewhat limited.
The manuscript lacks of a clear organizational structure, making it difficult for readers to follow the main arguments and conclusions. A more coherent structure, with distinct sections for different types of stem cells and specific effects of anaesthetics, could improve readability and comprehension.
Comments on the Quality of English LanguageModerate editing is required.
Author Response
Dear reviewer:
We sincerely appreciate the time you took to provide us with such valuable criticism and suggestion. It was a huge help in improving this article! We attach great importance to this, and through a more complete literature search and comprehensive analysis, this review has been comprehensively and deeply revised. We're so grateful for your input and want to make sure we address everything you've mentioned. Here are our point-by-point replies.
Q1: This review does not provide new insights or a fresh perspective on the impact of inhaled anaesthetics on stem cells. In fact literature is plenty of papers on this very controversial topic. The authors primarily summarise existing literature without offering a unique angle or novel hypotheses, which is crucial for a review paper to contribute meaningfully to the field.
A1: In response to the novel insights and hypotheses regarding inhaled anesthetics and stem cells, we have added a new section in the revision focusing on several key aspects: (1) the limitations of clinical research and the future studies of the novel inhalation anesthetic xenon; (2) recent advances and future trends in the effects of inhaled anesthetics on stem cell phenotypes; (3) clinical application prospect of inhaled anesthetics optimizing stem cells; (4) personalized application of inhalation anesthetics and ideas of transformation into clinical practice. Please refer to the section “6. Conclusions and Future Perspectives” in the revised manuscript for more details. Thank you!
Q2: The discussion does not delve deeply enough into the mechanisms by which anesthetics affect stem cells. While the paper covers various signaling pathways and molecular interactions, it often stops short of critically evaluating conflicting findings or identifying gaps in current knowledge.
A2: Thank you for your valuable comments! In the first draft, various molecular regulatory mechanisms were listed. Here, we focus on categorizing the isolated molecular mechanisms into four main areas related to stem cells: cell cycle, metabolism, stemness, and niche. Among them, metabolism and niche are the more popular stem cell research fields at present. This restructuring contributes to a better understanding of the different molecular mechanisms and how they regulate each other and improves readability for readers. In addition, four new mechanism diagrams (Figure 1-4) were drawn. For further details, please refer to the section “5. Mechanisms of inhalation anesthetics on stem cells" in the revised manuscript. Thank you for your insightful comments, which have guided us in improving our discussion.
As your suggestion “it often stops short of critically evaluating conflicting findings or identifying gaps in current knowledge”, we delved into the results of different mechanisms and highlighted conflicting findings, such as the role of microglia on NSCs and p-ERK on the cell cycle of stem cells.
“ Studies have shown that Iso and Sev induce neurodevelopmental toxicity by regulating microglial activation and increasing levels of pro-inflammatory cytokines (e.g., IL-6 and TNF-α), resulting in learning and memory impairments in neonatal rats[38]. However, microglia are also capable of releasing factors that support NSCs proliferation and maintenance, indicating their dual role in regulating stem cell behavior[133].”( Page 12, line 487 in the revised manuscript)
“Furthermore, Chen et al. discovered that overexpression of Hsp70 in NSCs can prevent the decline in cognitive abilities observed in offspring mice exposed to Sev during pregnancy[72]. Previous studies have shown that Hsp70 can activate extracellular signal-regulated kinase (ERK) related signaling pathways, which are critical in regulating cell proliferation and differentiation[73]. Activated mitogen-activated protein kinase (MAPK) -ERK can induce CyclinD expression and promote cell G1/S transition[74,75]. Therefore, it is plausible to hypothesize that Hsp70 modulates the cell cycle in NSCs through the activation of ERK-related signaling pathways, thereby mitigating the toxic effects of Sev on these cells. Gamma-aminobutyric acid (GABA) is an inhibitory neurotransmitter present in various cell types within the central nervous system. It plays a critical role in regulating cellular proliferation, differentiation, immune modulation, and other physiological processes[76-78]. 4.1% Sev treatment of ESCs cells for 6 h, p-ERK phosphorylation level increased, cell cycle arrest in S phase, self-renewal was inhibited, stem cell gene expression level decreased, proliferation slowed down, apoptosis increased. Knockdown of γ-aminobutyric acid A receptor (GABAAR) attenuates Sev promotion of p-ERK phosphorylation and rescues ESCs from Sev effects[43]. Activation of ERK is generally thought to contribute to cell proliferation[79], however, recent studies have shown that ERK may also have a pro-apoptotic effect [16]. Different cell types and different processing conditions may be responsible for this contradictory result.”( Page 7, line 306 in the revised manuscript)
Q3: There seems to be a lack of emphasis on the most recent advances and emerging trends in the field. A comprehensive review should highlight the latest studies and how they contribute to or challenge existing paradigms.
A3: We agree! In accordance with your recommendation, we conducted an analysis and comparison of the extant literature, which revealed that, in recent years, the influence of inhaled anesthetic on stem cells is no longer limited to proliferation and differentiation phenotypes, but more attention is paid to stem cell migration. In the section of “6. Conclusions and Future Perspectives”, we added relevant content and identified the emerging trends.
“Previous studies investigating the effects of inhalation anesthetics on stem cells have concentrated on the biological processes of proliferation and differentiation. In recent years, the studies of inhaled anesthetics on stem cell migration have received much attention[145,146]. Cell migration is fundamental for multicellular organisms to establish and maintain normal tissue[147]. The inherent migratory capacity of stem cells aids in maintaining tissue homeostasis and promoting repair and regeneration[148]. Extracellular vesicles are a class of cell-derived membrane structures, including exosomes and microvesicles, involved in various physiological and pathological processes. Extracellular vesicles are now recognized as mechanisms of intercellular communication, allowing cells to exchange proteins, lipids, and genetic material[149]. Stem cells secrete protective factors through extracellular vesicles to create a regenerative microenvironment that facilitates tissue maintenance and repair processes[150]. However, the effects of inhalation anesthesia on extracellular vesicles of stem cells have not been reported. In the future, priority should be given to extracellular vesicle secretion of stem cell, which is expected to make significant progress[151]. ”(Page 14, line 539 in the revised manuscript)
Q4: The review does not adequately address the limitations of the studies it cites, nor does it discuss the controversies or debates within the field. A critical evaluation of methodologies and findings is necessary to provide a balanced and informative overview.
A4: We acknowledge this issue and have perfected the manuscript to emphasize the discussion of the contradictory conclusions. First, we highlight the conflicting results obtained at different concentrations and treatment times for the concentration and time dependence of inhaled anesthetics and analyze possible influencing factors. The contents are as follows.
“Additionally, it is important to acknowledge that the sources of these NSCs and their cultural conditions are not entirely uniform, and these subtle differences may also influence the conclusions drawn from the experiments. Therefore, further rigorous and comparable research is needed to address the issues related to concentration and time dependency.”(Page 3, line 124 in the revised manuscript)
Moreover, we note that current research is primarily limited to in vitro cell studies, with a few in vivo investigations yielding results that differ from those observed in vitro. We discussed it as follows.
“While the diminished proliferation capacity of NSCs in vitro can recover within 24 hours post-exposure to 0.7% Iso, the effects on NSCs in vivo may persist for days or even weeks. Iso induces proliferation inhibition by causing cell cycle arrest directly in NSCs. The enduring anti-proliferative effects observed in vivo may be due to the secondary impacts of Iso on peripheral nervous tissue[31]. Additionally, Nie et al. noted that brief exposure (1 hour) to clinical levels of Sev enhanced NSCs proliferation in vitro, yet did not stimulate neurogenesis in vivo[30]. Discrepancies between in vivo and in vitro outcomes suggest that the regulation of stem cell proliferation in vivo is markedly more complex. It is reported that the fate of NSCs is precisely controlled by alterations in their microenvironment, referred to as the niche. The self-renewal and differentiation of NSCs rely on their interactions within this microenvironment. For instance, cerebrospinal fluid fosters NSCs proliferation in the subventricular zone by way of epithelial sodium channels, while angiogenesis in the niche is intricately linked to neurogenesis[34]. Following exposure to inhaled anesthetics, alterations may occur in neurotransmitters, growth factors, and membrane-associated ligands[35,36]. These changes may collectively affect the behavior of NSCs. Actually, in the clinical setting, the mechanisms are often intertwined. Hence, further research is necessary to accurately delineate the effects of inhalation anesthetics on NSCs in vivo. ”(Page 4, line 137 in the revised manuscript)
And, the studies of the effects of inhaled anesthetic on stem cells are limited to cell and animal experiments, which is undoubtedly an obvious limitation. We added the following in the revised manuscript.
“Numerous studies have investigated the effects of inhalation anesthetics on stem cells and their underlying mechanisms through animal models and cell experiments. However, due to safety and technical limitations, few clinical studies examining the effects of inhaled anesthetics on human stem cells have been reported. With the continuous advancement of new technologies such as in vivo imaging and single-cell sequencing, there is hope for conducting more clinical studies on humans in the near future. These studies aim to provide direct evidence validating the effects of inhalational anesthetics on human stem cells.” (Page 13, line 526 in the revised manuscript)
Q5: The manuscript focuses extensively on certain signaling pathways, such as the Akt-GSK-3β pathway, potentially overlooking other important mechanisms or broader implications. A more balanced discussion covering a wider range of molecular mechanisms would provide a more holistic view.
A5: Considering your suggestion, we abandoned the original mechanism of listing signaling pathways and molecules, and integrated the molecular mechanisms according to the four categories of cell cycle, metabolism, stemness, and niche. The enhanced discussion will help clarify the multiple layers of interaction that govern the effects of these anesthetics on stem cells, ultimately contributing to a deeper understanding of their action mechanisms. Thank you for the insightful suggestion.
Q6: While the paper mentions the implications for personalized medicine, it does not sufficiently explore how the findings could be translated into clinical practice. Discussions on potential therapeutic applications, safety concerns, and future research directions are somewhat limited.
A6: We appreciate your suggestions and have made sure to incorporate them into the manuscript. In the “6. Conclusions and Future Perspectives” section of the revised manuscript, we fully explore the individualized application of inhalation anesthetics on stem cells, including the effects of inhalation anesthetics on individuals of different ages, different types of tumors, and different types of stem cells. The contents are as follows.
“We also observe that the application of the same inhalational anesthetic can lead to neuroprotection in pediatric patients while causing neuronal damage in the elderly[157,158]. Animal studies have demonstrated that under identical conditions, treatment with Iso results in memory impairments exclusively in juvenile rats and not in adult ones[159]. This difference suggests that inhaled anesthetics may play different roles at different ages. Furthermore, the impact of inhalation anesthetics may vary among different types of cancer. For instance, Sev has been observed to inhibit the self-renewal of CD34+ stem/progenitor cells in chronic myeloid leukemia, while simultaneously promoting the proliferation of GSCs[55,62]. Additionally, the same inhalational anesthetic may elicit markedly different responses in various cell types. For example, Sev can enhance the proliferation of GSCs while suppressing the proliferation of NSCs[54,107]. These discrepancies may arise from variations in underlying mechanisms. With the rapid advancement of high-throughput sequencing technologies, we can develop strategies to compare molecular expression differences among various stem cells treated with inhalation anesthetics, thereby analyzing the specific mechanisms underlying their differential effects. This could facilitate a better understanding of how inhalation anesthetics operate across different stem cell types. What is intriguing is whether the mechanism by which inhaled anesthetic inhibits the proliferation of NSCs is applicable to the inhibition of tumor progression by CSCs. Conversely, does the mechanism that promotes GSCs proliferation also support NSCs self-renewal? Therefore, determining how to apply tailored treatment protocols for different patient populations while ensuring safety to maximize the therapeutic potential of inhalation anesthetics remains an area worthy of further investigation.”(Page 14, line 569 in the revised manuscript)
In addition, we highlight the importance of safety issues for specific patient groups and discuss potential therapeutic applications and future research directions. They are described in detail in the sections “3.ESCs” and “4.CSCs” and “6. Conclusions and Future Perspectives ”.
“Inhalation anesthetics are thought to be harmful rather than protective to ESCs, in contrast to the bidirectional effect of inhalation anesthetics on NSCs. It is crucial to investigate effective strategies for mitigating the adverse effects of inhaled anesthetics on embryonic development, including abortion and developmental delay. Furthermore, additional research is needed to determine whether inhalation anesthetics could potentially exhibit any protective effects on ESCs. This will provide important insights for developing safer and more effective treatment strategies for anesthesia during pregnancy.” ( Page 5, line 211 in the revised manuscript)
“Both clinical and basic studies have found a variety of effects of inhalation anesthetics on cancer or CSCs. The effects of inhalation anesthetics on CSCs may be influenced by numerous factors, including the concentration of inhaled anesthetics, the duration of treatment, and variations in the source of cells utilized. To obtain more precise and dependable conclusions, further research is essential to facilitate the judicious use of inhaled anesthetics for more effective inhibition of cancer progression and recurrence.” ( Page 6, line 261 in the revised manuscript)
“Moreover, the effects of inhalation anesthetics on stem cells provide new perspectives for clinical applications, especially in the realm of regenerative medicine. Regenerative medical technologies centered around stem cells hold immense potential for improving human diseases, delaying aging, and enabling tissue regeneration[152-154]. One of the major challenges facing stem cell transplantation is immune rejection, which leads to the transient survival of transplanted stem cells. Preconditioning with inhalation anesthetics prior to transplantation might enhance the survival and self-renewal of stem cells in vivo. Additionally, the primary methods of stem cell transplantation involve non-invasive blood transfusion and invasive targeted injections, with the latter requiring surgical intervention[155]. Prior research has demonstrated that inhalation anesthetics can mitigate the perioperative immune response via a range of molecular mechanisms[156]. It would be of interest to investigate whether the appropriate selection of inhalation anesthetics during surgery can affect the microenvironment of stem cells, reduce immune rejection, and improve the survival rate of transplanted stem cells.” (Page 14, line 554 in the revised manuscript)
Q7: The manuscript lacks of a clear organizational structure, making it difficult for readers to follow the main arguments and conclusions. A more coherent structure, with distinct sections for different types of stem cells and specific effects of anaesthetics, could improve readability and comprehension.
A7: In the revised manuscript, we described the effects of inhalation anesthetics on three types of stem cells separately, and elucidated the relevant molecular mechanism from four aspects including cell cycle, metabolism, stemness, and niche. We hope the regulation of organizational structure could improve readability and comprehension.
For a more detailed overview, we would like to kindly refer you to the revised manuscript. Thank you so much!
Reviewer 2 Report
Comments and Suggestions for Authors
An interesting manuscript has been submitted. Hao et al. collected information regarding the effect of inhaled anesthetics on the functions of different types of stem cells, such as neural, embryonic, and cancer stem cells. The authors systemically reviewed the molecular pathways of each cell type and elucidated how inhalation anesthetics affect them. The topic and title of this manuscript seem to be on point. This manuscript can be considered for publication after addressing the following minor comments.
1. It seems that references are missing in many paragraphs. For example, only one reference is cited in sections 2.3.5, 3.2, and 3.3. Does that mean that the information in the entire section comes from a single source?
2. The authors mix American English with British English (Ex, behavior and behaviour). Please unify it.
3. (Page 4, line 171) Research cannot be plural.
4. (Page 5, line 183 - 185) Neural stem cell should be abbreviated.
5. (Page 5, line 185 - 187) The analysis of the two contrast studies (line 181 – 185) is somewhat ambiguous. An explicit explanation seems to be required.
6. (Page 11, line 437) “This, It is therefore –” An awkward expression, and “I” should not be capitalized.
Author Response
Dear reviewer:
We sincerely thanks you for your feedback which help to improve the quality of our manuscript. Based on your suggestion, we have made corrected modifications to the manuscript. We try our best to supplement the related literature. And, we are very sorry for our incorrect writing. Here are our replies.
Q1: It seems that references are missing in many paragraphs. For example, only one reference is cited in sections 2.3.5, 3.2, and 3.3. Does that mean that the information in the entire section comes from a single source?
A1: It is really true as you suggested that we should cite more references. We have supplemented the above into the revised manuscript, as follows.
“p53 functions as a transcription factor that activates a wide array of genes, ultimately inducing apoptosis and cell cycle arrest[68]. p21 is one of the transcriptional targets of p53, and mediates p53-induced G1 cell cycle arrest[69]. Following treatment with varying concentrations of Iso, the liver kinase B1(LKB1) phosphorylates the p53 transcription factor at specific sites, enhancing its affinity for its target promoter, the p21 gene. Increased expression of p21 subsequently affects the formation of Cdks and cyclin complexes, leading to cell cycle arrest[70,71]. ” (Page 7, line 299 in the revised manuscript)
“Gamma-aminobutyric acid (GABA) is an inhibitory neurotransmitter present in various cell types within the central nervous system. It plays a critical role in regulating cellular proliferation, differentiation, immune modulation, and other physiological processes[76-78]. 4.1% Sev treatment of ESCs cells for 6 h, p-ERK phosphorylation level increased, cell cycle arrest in S phase, self-renewal was inhibited, stem cell gene expression level decreased, proliferation slowed down, apoptosis increased. Knockdown of γ-aminobutyric acid A receptor (GABAAR) attenuates Sev promotion of p-ERK phosphorylation and rescues ESCs from Sev effects[43].”(Page 7, line 314 in the revised manuscript)
“Retinoic acid receptor gamma (RAR-γ) plays a critical role in balancing the processes of stem cell self-renewal and differentiation[114,115]. Low expression or loss of RAR-γ results in a decrease in the self-renewal capability of stem cells, accompanied by an increase in differentiation. Research has shown that Iso inhibits the transcription of stemness factors Nanog and Oct4 by downregulating RAR-γ, thus hindering the self-renewal of mouse ESCs[44].” (Page 11, line 433 in the revised manuscript)
Q2, 3, 4, 6: The authors mix American English with British English (Ex, behavior and behaviour). Please unify it.
(Page 4, line 171) Research cannot be plural.
(Page 5, line 183 - 185) Neural stem cell should be abbreviated.
(Page 11, line 437) “This, It is therefore –” An awkward expression, and “I” should not be capitalized.
A2, 3, 4, 6: We sincerely apologize for any confusion caused by the differences between American and British English, as well as the issues related to singular and plural forms and phrase abbreviations. A comprehensive examination of the issue has been conducted, and the grammar and vocabulary in the full text have been duly rectified. We are grateful for your assistance in identifying and rectifying these errors.
Q5: (Page 5, line 185 - 187) The analysis of the two contrast studies (line 181 – 185) is somewhat ambiguous. An explicit explanation seems to be required.
A5: Sorry for the inconvenience in your reading due to the confusion in the writing. It has been demonstrated that viable microglia can release factors that promote the proliferation of NSCs. However, sevoflurane and isoflurane have been shown to inhibit the proliferation of NSCs by activating microglia. This evidence suggests that microglia exert an influence on the biological behavior of NSCs through more complex pathways. In the revised manuscript, the text was reformulated as follows:
“Studies have shown that Iso and Sev induce neurodevelopmental toxicity by regulating microglial activation and increasing levels of pro-inflammatory cytokines (e.g., IL-6 and TNF-α), resulting in learning and memory impairments in neonatal rats[38]. However, microglia are also capable of releasing factors that support NSCs proliferation and maintenance, indicating their dual role in regulating stem cell behavior[133].” (Page 12, line 487 in the revised manuscript)
Thank you very much for your attention and time!
Reviewer 3 Report
Comments and Suggestions for Authors
Thank you for permitting me to review this manuscript
In this paper the authors adress the effect of inhalational aenthetics on different stem cells and conclude on disruption on regulation of different types of stem cells
I have some minor suggestions
line 104 please provide reference (PPR)
Figure 1 and 2 need more text legends . In addition since only sev and Iso are displayed the title should only cite these 2 agents only
It should be emphacized that in clinical situations for humans high concentration exposure (like sevo up to 4% is often less than few minutes , but the main qusetion remain as what about the longer expoure to clinical concentration such as 2% sevoflurane or other
In addition naturallu all og these experiment were conducted in animals and the real effect iin humans are not verified
conclusion should be rewritten in a clear manner, for ex there is no necessity to remember again the role of stem cells , and most importantly avoid mixed "bring home" messages
Author Response
Dear reviewer:
Thank you for reviewing our paper and providing your valuable comments. We greatly appreciate the time you took to carefully read our paper and provide constructive feedback. Based on your suggestions, we have made the appropriate revisions and additions to the paper. Below are our answers to your questions:
Q1: Figure 1 and 2 need more text legends. In addition since only sev and Iso are displayed the title should only cite these 2 agents only.
A1: Thank you for your suggestions. The new mechanism diagrams were drawn, and the related figure legends were added. Please refer to Figures 1-4 in the revised manuscript.
Q2: It should be emphacized that in clinical situations for humans high concentration exposure (like sevo up to 4% is often less than few minutes , but the main qusetion remain as what about the longer expoure to clinical concentration such as 2% sevoflurane or other.
A2: We fully agree with you. It is exactly what we care about. It is common practice to utilize lower concentrations of inhaled anesthetics for the maintenance of clinical anesthesia. In the revised manuscript, we tried our best to search and supplement the studies with clinical doses of inhaled anesthetics affecting stem cells as following.
“In vivo experiments further validated these conclusions. Pregnant mice were treated daily with 1.4% Iso for 2 hours over three consecutive days during gestation days E3.5 to E6.5, simulating the exposure conditions experienced by pregnant women undergoing prolonged non-obstetric surgeries. At E18, the pregnant mice were euthanized, and the weights of their pups were measured, revealing impaired fetal growth and development[44,45].” (Page 5, line 198 in the revised manuscript)
“Treatment of glioma stem cells (GSCs) derived from high-grade glioma patients with 2% Sev (the minimum alveolar concentration during anesthesia in a clinical setting at the start of surgery) does not affect cell apoptosis, proliferation capacity, or colony formation ability, even after prolonged exposure (4-6 days)[53]. ” (Page 6, line 244 in the revised manuscript)
Q3: In addition naturallu all og these experiment were conducted in animals and the real effect in humans are not verified.
A3: As you said, there is a substantial body of research examining the effects of inhaled anesthetics on stem cells, particularly neural stem cells. However, it is important to note that all of these experiments have been conducted using animal or cell models. While these experiments have yielded valuable insights, they do not fully capture the complexity of human physiology. We discussed it in the revised manuscript.
“However, due to safety and technical limitations, few clinical studies examining the effects of inhaled anesthetics on human stem cells have been reported. With the continuous advancement of new technologies such as in vivo imaging and single-cell sequencing, there is hope for conducting more clinical studies on humans in the near future. These studies aim to provide direct evidence validating the effects of inhalational anesthetics on human stem cells.”(Page 13, line 528 in the revised manuscript)
We are grateful to you for drawing attention to this crucial issue.
Q4: conclusion should be rewritten in a clear manner, for ex there is no necessity to remember again the role of stem cells , and most importantly avoid mixed "bring home" messages
A4: Thank you for your pertinent suggestion. We removed the duplication and ambiguity in the conclusion section and made a few adjustments and optimizations to the content.
“Taken together, further research is required to elucidate the long-term effects of inhalation anesthetics on stem cells and the precise regulation of stem cell behavior. These findings could help enhance precision anesthesia management and explore other clinical applications of inhaled anesthetics.” (Page 15, line 592 in the revised manuscript)
For a more detailed overview, we would like to kindly refer you to the revised manuscript. Thank you so much!
Round 2
Reviewer 1 Report
Comments and Suggestions for Authors
The Authors have answered to the previous concerns ameliorating the paper that now is improved.
Comments on the Quality of English LanguageMinor editing of English language required.